# Myocardial Infarction with Obstructive, Non-Obstructive, and Mimicking Conditions: Clinical Phenotypes, Diagnostic Imaging, Management, and Prognosis

**DOI:** 10.3390/jcm14093006

**Published:** 2025-04-26

**Authors:** Athanasios Samaras, Dimitrios V. Moysidis, Andreas S. Papazoglou, Georgios P. Rampidis, Konstantinos Barmpagiannos, Antonios Barmpagiannos, Christos Kalimanis, Efstratios Karagiannidis, Barbara Fyntanidou, George Kassimis, Apostolos Tzikas, Antonios Ziakas, Nikolaos Fragakis, Konstantinos Kouskouras, Vassilios Vassilikos, George Giannakoulas

**Affiliations:** 1Second Cardiology Department, Hippokrateion University Hospital, Aristotle University of Thessaloniki, 54124 Thessaloniki, Greece; ath.samaras.as@gmail.com (A.S.); stratoskarag@gmail.com (E.K.); gksup@yahoo.gr (G.K.); aptzikas@yahoo.com (A.T.); fragakis.nikos@googlemail.com (N.F.); 2Medical School, Aristotle University of Thessaloniki, 54124 Thessaloniki, Greece; dimoysidis@gmail.com (D.V.M.); anpapazoglou@yahoo.com (A.S.P.); grampidi@gmail.com (G.P.R.); kostasmparmp@hotmail.gr (K.B.); antonismparmpagiannos@hotmail.com (A.B.); kalimanisxristos@gmail.com (C.K.); 3Department of Emergency Medicine, AHEPA University Hospital, 54636 Thessaloniki, Greece; fyntanidou@auth.gr; 4First Department of Cardiology, AHEPA University Hospital, School of Medicine, Faculty of Health Sciences, Aristotle University of Thessaloniki, 54636 Thessaloniki, Greece; tonyziakas@hotmail.com; 5Department of Radiology, University General Hospital of Thessaloniki AHEPA, 54636 Thessaloniki, Greece; coskou@auth.gr; 6Third Cardiology Department, Hippokrateion University Hospital, Aristotle University of Thessaloniki, 54124 Thessaloniki, Greece; vvassil@auth.gr

**Keywords:** myocardial infarction with non-obstructive coronary artery disease, myocardial infarction mimickers, cardiac magnetic resonance, coronary computed tomography angiography, prognosis

## Abstract

**Background/Objectives:** Myocardial infarction with non-obstructive coronary arteries (MINOCA) is a heterogenous clinical entity that differs in pathophysiology, treatment, and prognosis from myocardial infarction with obstructive coronary artery disease (MI-CAD) and MINOCA mimickers, such as myocarditis or Takotsubo syndrome. This study aimed to compare the clinical characteristics, imaging findings, management strategies, and long-term outcomes of patients with true MINOCA, MI-CAD, and MINOCA mimickers. **Methods:** This retrospective cohort study included 1596 patients hospitalized with acute myocardial infarction (AMI) between 2012 and 2024 at a tertiary university hospital. Patients were classified as having true MINOCA, MI-CAD, or MINOCA mimickers based on coronary angiography and advanced cardiac imaging. Data included clinical and laboratory variables, echocardiography, cardiac magnetic resonance (CMR), and coronary computed tomography angiography (CCTA). All-cause mortality was analyzed using Cox regression. **Results:** Of 1596 AMI patients, 111 (7.0%) had true MINOCA, 1359 (85.1%) had MI-CAD, and 127 (8.0%) had MINOCA mimickers. Mimicker patients were significantly younger and had fewer cardiovascular risk factors. True MINOCA was more frequent in females and associated with preserved left ventricular ejection fraction and lower high-sensitivity troponin T levels compared to MI-CAD. CMR and CCTA contributed to etiological clarification in over 70% of MINOCA and mimicker patients. High-risk plaque features were observed in 42.9% of CCTA scans, suggesting but not confirming an atherosclerotic mechanism. Long-term all-cause mortality in MINOCA was similar to MI-CAD (32.1% vs. 30.9%, *p* = 0.764) and significantly higher than in mimickers (5.9%, *p* < 0.001). **Conclusions:** True MINOCA is a distinct clinical entity with diagnostic and prognostic implications. Its comparable mortality to MI-CAD highlights the need for accurate diagnosis and targeted secondary prevention strategies.

## 1. Introduction

Myocardial infarction with non-obstructive coronary arteries (MINOCA) is a diverse clinical condition characterized by the presence of myocardial infarction (MI) without significant obstructive coronary artery disease (CAD). In contrast to conventional myocardial infarction, primarily caused by plaque rupture and thrombus formation in major coronary arteries, MINOCA involves a variety of mechanisms, such as coronary vasospasm, microvascular dysfunction, spontaneous coronary artery dissection, and thrombosis with spontaneous lysis [1].

Despite advancements in cardiac imaging and biomarker assessment, the exact pathophysiological mechanisms of MINOCA remain incompletely understood, contributing to diagnostic uncertainty and therapeutic challenges [2]. Distinguishing MINOCA from MINOCA mimickers—conditions that present similarly but lack true myocardial infarction, such as myocarditis and Takotsubo cardiomyopathy—is critical for appropriate management [3]. Furthermore, understanding how MINOCA differs from MI with obstructive CAD (MI-CAD) is essential, as the latter follows well-established pathophysiological mechanisms with standardized treatment strategies, while MINOCA or MINOCA mimickers require a more nuanced diagnostic and therapeutic approach [4].

Contemporary research studies suggest that MINOCA or MINOCA mimickers may arise in up to 5–10% of the total MI population [5,6,7,8] and may be linked with a comparable prognosis with MI-CAD [9]. Given the diverse etiologies underlying MINOCA and MINOCA mimickers along with their complex prognostic course, a systematic approach integrating multimodal imaging, laboratory markers, and clinical risk stratification is crucial for improving patient management. This real-world study aimed to add to the existing literature our diagnostic, therapeutic, and prognostic experience among a well-characterized MI cohort, aiming to enhance the characterization of MINOCA and ultimately contribute to personalized therapeutic approaches.

## 2. Materials and Methods

### 2.1. Study Design and Population

This was a retrospective, observational cohort study conducted at the First Cardiology Department of AHEPA University Hospital in Thessaloniki (Greece), including consecutive patients admitted with an initial diagnosis of acute myocardial infarction between 2012 and 2024. Data were extracted from the MINOCA-GR [10] and CardioMining Databases [11]. The final cohort was stratified into the following three groups based on diagnostic evaluation: 1: true MINOCA, 2: MI-CAD, and 3: MINOCA mimickers, defined as patients presenting with an initial working diagnosis of MINOCA but ultimately diagnosed with a non-ischemic condition, including Takotsubo syndrome or myocarditis. All patients classified as “true MINOCA” or “MI-CAD” underwent coronary angiography during index hospitalization. Patients classified as “MINOCA mimickers” underwent angiography per physician preference.

Inclusion criteria were age ≥18 years and hospitalization for acute MI. Exclusion criteria included patients that died during hospitalization, and thus there is no available discharge data. Diagnosis of MI was based on the Fourth Universal Definition, requiring detection of a rise and/or fall in cardiac biomarkers with at least one value above the 99th percentile upper reference limit, accompanied by evidence of myocardial ischemia (clinical symptoms, ECG changes, or imaging findings).

### 2.2. Classification of Groups

True MINOCA was defined according to the 2019 American Heart Association scientific statement, requiring the presence of acute myocardial infarction, non-obstructive coronary arteries (<50% stenosis in any major epicardial vessel), and the exclusion of alternative diagnoses such as myocarditis or Takotsubo syndrome.

MI-CAD included patients with myocardial infarction and obstructive CAD, defined as ≥50% stenosis in at least one major coronary artery.

MINOCA mimickers included patients who initially fulfilled criteria for MINOCA (acute MI with non-obstructive coronary arteries) but were subsequently diagnosed with non-ischemic etiologies based on cardiac magnetic resonance imaging or other diagnostic tools. These comprised Takotsubo syndrome and myocarditis.

### 2.3. Data Collection

Demographic characteristics, presenting symptoms, cardiovascular risk factors, medical history, laboratory findings at admission, electrocardiographic (ECG) features, echocardiographic parameters, diagnostic imaging studies, in-hospital management, and discharge medication were extracted from electronic medical records.

### 2.4. Diagnostic Imaging

Cardiac magnetic resonance (CMR) was performed in a subset of patients with non-obstructive coronary arteries to evaluate for myocarditis, Takotsubo syndrome, infarction, or other structural abnormalities. The diagnosis of myocarditis was based on Lake Louise criteria, requiring the presence of myocardial edema and non-ischemic late gadolinium enhancement. Takotsubo syndrome was diagnosed based on established criteria, including transient wall motion abnormalities extending beyond a single coronary territory, absence of obstructive coronary disease or plaque rupture, new ECG abnormalities or modest troponin elevation, and exclusion of other causes.

Other imaging modalities, including coronary computed tomography angiography (CCTA), computed tomography pulmonary angiography (CTPA), single-photon emission computed tomography (SPECT), and ambulatory ECG (Holter) monitoring, were performed at the discretion of the treating physicians.

### 2.5. Outcomes

The clinical outcome of interest was all-cause mortality. Mortality data were obtained from hospital records or national death registries.

### 2.6. Statistical Analysis

Continuous variables were assessed for normality using the Shapiro–Wilk test. Variables with non-normal distribution were summarized as medians with interquartile ranges (IQR), and comparisons across the three groups were performed using the Kruskal–Wallis H test. Categorical variables were expressed as frequencies and percentages and compared using the chi-square test or Fisher’s exact test, as appropriate. A two-sided *p*-value of <0.05 was considered statistically significant. All statistical analyses were performed using R (version 4.4.2) and SPSS (version 28).

### 2.7. Ethical Considerations

This study was conducted in accordance with the Declaration of Helsinki and was approved by the appropriate Institutional Review Boards or Ethics Committees. All patient data were anonymized and de-identified prior to analysis to ensure confidentiality. No personal patient information was collected or stored. Due to the retrospective design, the requirement for individual patient consent was waived by the ethics committee in compliance with Greek regulatory guidelines.

## 3. Results

### 3.1. Study Population and Group Distribution

A total of 1596 patients were hospitalized for acute MI in our Cardiology Department from 2012 until 2024 (median year of hospitalization: 2017). Among these, 111 (7.0%) were classified as true MINOCA, 1359 (85%) as MI-CAD, and 127 (8.0%) as MINOCA mimickers (Table 1).

### 3.2. Clinical Profile of True MINOCA Patients

The median age of patients with MINOCA was 63 years, and approximately half were male. The median duration of hospitalization was 6 days. Most MINOCA cases presented as non-ST elevation myocardial infarction, with chest pain being the predominant symptom. Common cardiovascular risk factors included smoking in nearly half of the patients, arterial hypertension in over 40%, and dyslipidemia in almost 30%. Electrocardiographic abnormalities comprised mainly negative T waves in nearly 40% of patients. Discharge medications were largely guideline-directed, with antiplatelets used in over 93%, beta-blockers in 75%, and renin–angiotensin–aldosterone system inhibitors in just over half of the patients. However, dual antiplatelet therapy was prescribed in only 7 out of 10 patients with true MINOCA. Imaging and laboratory evaluations revealed a preserved median left ventricular ejection fraction, normal chamber dimensions, and moderate elevations in cardiac biomarkers such as high-sensitivity troponin T and N-terminal pro-B-type natriuretic peptide. In-hospital management universally included coronary angiography, while additional imaging such as CMR and CCTA was performed in more than half of patients (Table 1).

### 3.3. Comparison of MINOCA with MI-CAD and MINOCA Mimickers

#### 3.3.1. Demographic and Clinical Characteristics

Patients with MINOCA mimickers were significantly younger compared to those with true MINOCA or MI-CAD (median age 38 vs. 63 and 64, respectively, *p* < 0.001). Male sex was less prevalent in the true MINOCA and MINOCA mimicker groups (53.2% and 67.7%, respectively) than in the MI-CAD group (77.5%, *p* < 0.001). Hospitalization duration did not differ significantly across groups.

ST-elevation myocardial infarction (STEMI) was observed in 59.0% of MI-CAD cases, 5.4% of true MINOCA, and 0.8% of mimickers (*p* < 0.001). NSTEMI presentation was absent among mimickers but present in 55.0% of true MINOCA and 41.1% of MI-CAD (*p* < 0.001).

Chest pain was the predominant symptom at presentation in all groups. Fever was notably more common in MINOCA mimickers (28.3%) compared to other groups (*p* < 0.001). Dyspnea and palpitations showed no significant inter-group differences.

#### 3.3.2. Medical History and Cardiovascular Risk Factors

Smoking, hypertension, dyslipidemia, and diabetes mellitus were significantly more common in the MI-CAD group. Conversely, a history of myocarditis was exclusively observed in the MINOCA mimicker group (5.5%, *p* < 0.001). Chronic kidney disease was predominantly associated with MI-CAD (7.9%, *p* < 0.001). Connective tissue disease was more prevalent among mimickers and true MINOCA compared to MI-CAD (*p* = 0.035).

#### 3.3.3. Electrocardiographic Findings

ST-segment elevation at admission was most frequent in MI-CAD (39.7%) and lowest in true MINOCA (10.8%, *p* < 0.001). ST depression and pathological Q waves were similarly more common in MI-CAD. Negative T waves were significantly more prevalent in true MINOCA (39.6%) compared to MI-CAD (19.0%) and mimickers (22.0%, *p* < 0.001).

Although only one patient in the MINOCA mimicker group was initially classified as presenting with STEMI, ST-segment elevation on the admission electrocardiogram was documented in 34 patients (26.8%). This discrepancy reflects the fact that final diagnoses were reassigned based on imaging and clinical evolution, with several cases of Takotsubo syndrome and myocarditis presenting with ST-segment elevation but ultimately not fulfilling criteria for a true STEMI.

#### 3.3.4. Laboratory Values

Peak high-sensitivity troponin T levels were highest in MI-CAD (median 1314 [2963]), followed by mimickers (232 [766]) and true MINOCA (197 [699]) (*p* < 0.001), although the difference between the latter two groups was not statistically significant. White blood cell counts, blood glucose, creatinine, CPK, and NT-proBNP levels differed significantly across groups, with higher inflammatory and metabolic markers observed in MI-CAD.

#### 3.3.5. Echocardiographic Parameters

LVEF was higher in true MINOCA (59) and mimickers (60) compared to MI-CAD (52], *p* < 0.001). The E/E′ ratio was significantly lower in mimickers (6.3 [2]) than in MI-CAD and true MINOCA (*p* = 0.003). No significant differences were observed in left atrial volume or LVED diameter.

#### 3.3.6. Pharmaceutical Therapies

Antiplatelet therapy and dual antiplatelet therapy were predominantly used in MI-CAD (97.6% and 89.8%, respectively), while significantly less frequent in MINOCA mimickers (18.9% and 1.6%, respectively, *p* < 0.001). Beta-blockers, RAAS inhibitors, and statins were also more frequently prescribed in MI-CAD.

#### 3.3.7. Invasive Coronary Angiography

Invasive coronary angiographic findings were available for 171 patients with non-obstructive coronary arteries, including 117 with true MINOCA and 54 with MINOCA mimickers. Among these, 67 patients (39.2%) had completely normal coronary arteries, while 76 patients (44.4%) exhibited non-obstructive coronary artery disease with luminal irregularities or stenosis < 50%. Intermediate stenosis (50–70%) was identified in 16 patients (9.4%). Spontaneous coronary artery dissection was detected in 12 patients (7.0%) based on characteristic angiographic features. No cases of spontaneous vasospasm were observed, and pharmacologic provocation testing was not performed.

#### 3.3.8. CMR and CCTA Imaging

Advanced imaging modalities, particularly CMR and CCTA, were extensively utilized in patients with MINOCA and significantly contributed to diagnostic clarification. Among the 244 patients classified as either true MINOCA (*n* = 117) or MINOCA mimickers (*n* = 127), CMR was performed in 145 patients (59.4%) and CCTA in 72 patients (29.5%).

CMR was performed in 75 of 117 true MINOCA patients (64.1%), all of whom demonstrated findings consistent with an ischemic etiology. Specifically, all patients exhibited subendocardial or transmural late gadolinium enhancement (LGE) in a coronary distribution, confirming acute myocardial infarction. In the MINOCA mimicker group, CMR was performed in 61 of 127 patients (48.0%). Among these, 36 patients (59.0%) were diagnosed with myocarditis based on typical mid-wall or subepicardial LGE patterns, 21 patients (34.4%) had Takotsubo syndrome based on regional wall motion abnormalities and myocardial edema without LGE, and 4 patients (6.6%) had normal CMR findings. In contrast, CMR was used in only 9 of 1359 MI-CAD patients (0.7%) (*p* < 0.001).

CCTA was performed in 65 of 117 true MINOCA patients (55.5%), in 7 of 127 mimicker patients (5.5%), and in only 3 of 1359 MI-CAD patients (0.2%) (*p* < 0.001). Among true MINOCA cases, CCTA identified non-obstructive coronary atherosclerosis in 49 of 65 patients (75.4%). Notably, 21 of these 49 patients (42.9%) exhibited high-risk plaque features, including positive remodeling and low-attenuation plaque. In many of these cases, the location of the high-risk plaques anatomically corresponded with the infarct-related myocardial territory identified on CMR, reinforcing a diagnosis of type 1 myocardial infarction due to plaque disruption despite the absence of obstructive stenosis on invasive angiography. In the remaining 16 patients (24.6%), CCTA demonstrated completely normal coronary arteries.

Importantly, CCTA provided incremental value beyond CMR in 38 of 72 patients (52.8%) who underwent both modalities. In contrast, among mimicker patients, CCTA revealed completely normal coronary anatomy in 6 of 7 patients (85.7%).

In the MINOCA mimicker group (*n* = 127), the final diagnoses established through CMR, CCTA, and clinical adjudication were as follows: 36 patients (28.3%) were diagnosed with myocarditis, 21 patients (16.5%) with Takotsubo syndrome, 18 patients (14.2%) with pericarditis or perimyocarditis, 5 patients (3.9%) with pulmonary embolism, and 4 patients (3.1%) with type 2 myocardial infarction attributed to demand ischemia from non-coronary causes. Additionally, 3 patients (2.4%) were diagnosed with sepsis-associated myocardial injury or cytokine-mediated injury, and 4 patients (3.1%) had transient myocardial injury due to non-structural causes such as severe electrolyte abnormalities or hypertensive crises. In the remaining 36 patients (28.3%), advanced imaging failed to identify a specific myocardial or coronary pathology, and they were thus classified as having undetermined non-ischemic mechanisms. These patients likely experienced transient myocardial injury due to systemic conditions or physiological stressors without structural cardiac abnormalities.

In total, among the 244 patients with non-obstructive coronary arteries, advanced imaging with either CMR or CCTA was performed in 181 patients (74.2%), contributing to diagnostic clarification in 170 patients (69.7%).

#### 3.3.9. Outcomes

Follow-up data were available for 1434 patients (90% of the total) (Table 1). During a median follow-up of 6 years, 427 patients (26.8%) died from any cause. All-cause mortality was significantly higher among patients with MI-CAD (30.9%) and those with true MINOCA (32.1%) compared to patients with MINOCA mimickers (5.9%) (*p* < 0.001). Kaplan–Meier survival analysis (Figure 1) highlighted the most favorable long-term survival in the MINOCA mimics group, whereas patients with true MINOCA had the poorest survival trajectory over time (log-rank *p*-value = 0.002).

### 3.4. Diagnostic Predictors Between True MINOCA and MI-CAD

In multivariable logistic regression analysis evaluating distinct diagnostic features between true MINOCA vs. MI-CAD, five variables were independently associated with diagnostic classification (Table 2). Male sex and ST-segment elevation at admission were strongly associated with more than 3-fold odds of MI-CAD compared with MINOCA. Older age and higher admission levels of hs-TnT were also positively associated with obstructive MI. In contrast, preserved left ventricular ejection fraction (LVEF ≥ 50%) was associated with increased odds of true MINOCA (Figure 2).

## 4. Discussion

This real-world study, the first relevant cohort study in Greece, provided a comprehensive analysis of MINOCA, highlighting key etiological contributors and their prognostic implications. Our findings indicate that patients with true MINOCA exhibit distinct clinical and imaging characteristics compared to those with MI-CAD and MINOCA mimickers. True MINOCA accounted for 7% of MI cases, while MI-CAD remained the predominant etiology (85%). MINOCA mimickers (8% of the total population) were significantly younger compared to true MINOCA and MI-CAD, with a lower prevalence of cardiovascular risk factors. Patients with MINOCA appeared to have a distinct clinical phenotype, including fewer cardiovascular risk factors, lower biomarker levels, and more favorable echocardiographic parameters compared to those with obstructive CAD. However, long-term all-cause mortality in patients with true MINOCA was comparable to those with MI-CAD (32.1% vs. 30.9%) and significantly higher than in MINOCA mimickers (5.9%). Furthermore, advanced imaging modalities, including CMR and CCTA, were crucial for diagnostic clarification, with CMR distinguishing ischemic infarction from alternative diagnoses in 93% of MINOCA mimickers. High-risk plaque features were identified in 43% of true MINOCA patients using CCTA. Management strategies differed significantly, with dual antiplatelet therapy and PCI primarily used in MI-CAD, while conservative management was more common in MINOCA and mimicker cases.

### 4.1. Prevalence

MINOCA prevalence seems to vary significantly across studies due to differing definitions and diagnostic criteria [1]. For instance, a most recent study reported that among 8560 STEMI patients, 4.8% of them had non-obstructive CAD, including 1.4% with true MINOCA and 3.4% with MINOCA mimickers [8]. Other large cohort studies, published during the last 5 years, indicate a prevalence rate of 1% to 4% for MINOCA among acute MI patients [5,6,7,12]. Specific global prevalence data for MINOCA mimickers are limited, as their identification heavily relies on the availability and application of comprehensive diagnostic evaluations, including CMR [13]. 

### 4.2. Diagnostic Yield of CMR and CCTA in Suspected MINOCA

The diagnosis of MINOCA remains a challenge due to its heterogeneous pathophysiology, which includes coronary plaque disruption, coronary vasospasm, coronary embolism, microvascular dysfunction, and myocardial disorders such as Takotsubo syndrome or myocarditis [14]. Studies emphasize the importance of multimodal imaging, including coronary angiography with intravascular ultrasound (IVUS) or optical coherence tomography (OCT), to detect plaque rupture, erosion, or thrombus, which may not be visible on conventional angiography. A study by Reynolds et al. highlighted that nearly 40% of MINOCA patients had evidence of plaque disruption on IVUS, underscoring the role of intravascular imaging in identifying underlying mechanisms [15]. Another study demonstrated that OCT identified culprit lesions in 77.5% of MINOCA cases, with 12.5% of patients displaying multiple hyperenhanced myocardial lesions, suggesting coronary embolization as a potential mechanism [16]. Similarly, CMR has been shown to be a pivotal tool in distinguishing between ischemic and non-ischemic causes, such as myocarditis or stress cardiomyopathy, which can mimic MINOCA. CMR can detect infarct-related edema via T2-weighted imaging and identify fibrosis using LGE and, thereby, differentiate MINOCA from myocarditis, stress cardiomyopathy, or other non-ischemic causes [17]. Notably, the combined use of OCT and CMR has been shown to result in a diagnosis for 100% of patients classified as MINOCA, highlighting the powerful synergy between intracoronary and cardiac imaging in elucidating the underlying etiology [18]. The integration of OCT with CMR significantly enhances diagnostic accuracy, achieving a yield of 85% compared to 44% with OCT alone and 74% with CMR alone [18].

CCTA has emerged as a valuable non-invasive imaging modality in the diagnostic work-up of patients with suspected MINOCA, particularly in distinguishing ischemic mechanisms from non-ischemic mimickers [19]. CCTA offers high-resolution visualization of coronary anatomy, enabling the detection of non-obstructive atherosclerotic plaque that may be underestimated or missed by invasive coronary angiography [20]. Such plaques may represent the substrate for plaque rupture or erosion—mechanisms increasingly recognized in the pathophysiology of MINOCA [21]. In particular, CCTA can identify high-risk plaque features, including positive remodeling, low-attenuation plaque, and napkin-ring sign, all of which are associated with coronary instability even in the absence of significant luminal stenosis [22,23]. In studies involving MINOCA patients with infarction confirmed by cardiac magnetic resonance imaging (CMR), CCTA has uncovered culprit plaques not visualized on angiography, often located within the infarct-related artery and exhibiting expansive remodeling and larger plaque burden, supporting a diagnosis of type 1 myocardial infarction. Conversely, CCTA can confirm entirely normal coronary arteries in patients ultimately diagnosed with non-ischemic conditions such as Takotsubo syndrome or myocarditis. In this context, CCTA serves as a gatekeeper by excluding obstructive or high-risk coronary lesions, thereby justifying a reclassification of the clinical event. Additionally, although CCTA does not directly assess microvascular function, its ability to rule out significant epicardial coronary disease supports the use of downstream functional testing (e.g., myocardial perfusion imaging or invasive coronary physiology) when microvascular dysfunction is suspected [24]. Emerging CCTA-based tools, such as fractional flow reserve derived from CT (FFR-CT) and the pericoronary fat attenuation index (pFAI), are being investigated for their potential to identify functional ischemia or coronary inflammation [25,26]. Overall, CCTA complements CMR by elucidating the coronary substrate of MINOCA, refining diagnostic classification, and informing therapeutic decisions—whether through the identification of subclinical yet high-risk atherosclerosis requiring secondary prevention or by reinforcing a non-ischemic diagnosis when coronary arteries appear truly normal.

### 4.3. Prognosis and Management

The prognosis of patients with MINOCA depends on the underlying cause and is currently under active investigation [4]. While some studies suggest that MINOCA patients may have a better short-term prognosis compared to those with MI-CAD [5,12,27], the long-term outcomes are concerning, with significant risks of mortality and adverse cardiovascular events that might even exceed the risk of MI-CAD patients [7,12,28,29]. A meta-analysis of MINOCA studies demonstrated similar results, with a pooled in-hospital mortality rate of 0.9% but a pooled 12-month mortality rate of 4.7% [27]. The prognosis of MINOCA mimickers appears to be comparable to or slightly better than that of MINOCA patients [29], though further research is needed to elucidate these differences fully.

Since the underlying cause of MINOCA is identified, management strategies must be tailored accordingly. For cases where an ischemic mechanism is confirmed—such as plaque rupture, coronary spasm, or thrombus—treatment aligns with standard MI guidelines, similar to patients with MI-CAD. These individuals should receive secondary prevention strategies, including antiplatelet therapy, statins, ACE inhibitors, and beta-blockers, in line with established cardioprotective measures [4]. The American Heart Association specifically recommends that when plaque disruption or another ischemic etiology is detected, post-MI treatment should mirror that of MI-CAD [4]. However, real-world data suggest that patients with MINOCA often receive less intensive pharmacologic therapy at discharge compared to those with MI-CAD [30]. Registry data indicate that standard MI treatments such as aspirin, beta-blockers, and statins are prescribed less frequently to MINOCA patients, likely reflecting uncertainty in diagnosis or the heterogeneous nature of the condition [29]. Moreover, traditional post-discharge therapies used in acute MI, such as DAPT, appear to have a neutral prognostic effect in patients with a generic diagnosis of “MINOCA” [31]. These discrepancies emphasize the importance of precise diagnostic workup to guide appropriate management.

For conditions that mimic MINOCA but do not involve a primary ischemic mechanism, management diverges significantly [32,33]. In Takotsubo cardiomyopathy, the treatment is primarily supportive, focusing on hemodynamic stabilization and addressing potential complications such as heart failure or arrhythmias [34]. Long-term, beta-blockers have been considered for recurrence prevention, although evidence remains inconclusive regarding their efficacy [34]. Similarly, myocarditis management varies depending on the subtype. Importantly, in myocarditis, antithrombotic and ischemia-directed therapies are generally unwarranted unless there is another clear indication, distinguishing it from true ischemic injury. These key differences highlight the risk of broadly categorizing MINOCA mimickers with ischemic MINOCA, as a one-size-fits-all approach could lead to inappropriate treatment.

It is worth mentioning that in addition to established non-ischemic mimickers such as myocarditis and Takotsubo syndrome, clinicians should be aware of less common conditions that can produce ‘pseudo-ischemic’ electrocardiographic changes in patients presenting with chest pain. These include mitral valve prolapse, particularly in association with concave-shaped chest wall conformation, as well as hyperventilation-induced electrocardiographic abnormalities [35]. Both conditions have been associated with transient ST-segment or T-wave changes that may be erroneously interpreted as signs of acute ischemia, sometimes leading to unnecessary coronary angiography, even when performed by experienced cardiologists. Awareness of these mimickers is critical for avoiding misdiagnosis and preventing invasive testing in cases where structural and functional myocardial integrity is preserved.

## 5. Limitations

This study has some limitations that warrant consideration. First, its retrospective, single-center design may introduce selection bias and limit the generalizability of the findings to other healthcare settings or populations. Second, the classification of patients into true myocardial infarction with non-obstructive coronary arteries, myocardial infarction with obstructive coronary artery disease, and myocardial infarction mimickers relied on the availability and interpretation of advanced imaging, which was not systematically performed in all patients and may have led to misclassification in a subset of cases. Third, the absence of intracoronary imaging modalities such as optical coherence tomography or intravascular ultrasound during coronary angiography may have limited the identification of underlying mechanisms such as plaque rupture, erosion, or epicardial vasospasm in patients with angiographically non-obstructive disease [36]. Fourth, data on cause-specific mortality and non-fatal cardiovascular events were not uniformly available, precluding a more granular assessment of clinical outcomes. Lastly, although discharge medication use was reported, data on adherence, dose titration, and long-term medical management were not captured, which may have influenced the observed outcomes.

## 6. Conclusions

In this single-center cohort study, true MINOCA accounted for 7% of total acute MI cases and was associated with a distinct clinical profile in terms of traditional risk factors, biomarker levels, and cardiac function compared to MI-CAD. Despite these features, long-term all-cause mortality in patients with MINOCA was comparable to that of MI-CAD but significantly higher than that of patients with MINOCA mimickers. Advanced imaging with CMR and CCTA was essential for identifying myocardial injury patterns, detecting high-risk but non-obstructive coronary plaques suggestive of an ischemic mechanism, and distinguishing non-ischemic mimickers. However, definitive confirmation of atherosclerotic plaque disruption would require intracoronary imaging. These findings highlight the prognostic significance of true MINOCA and the need for comprehensive diagnostic evaluation to guide personalized management and improve long-term outcomes.

## Figures and Tables

**Figure 1 jcm-14-03006-f001:**
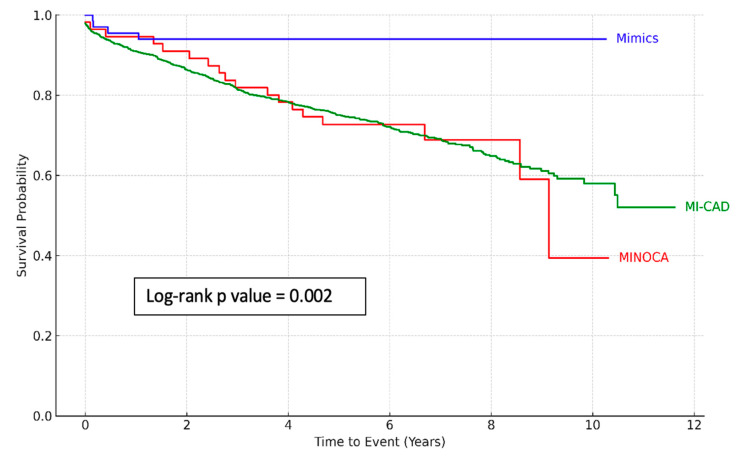
Kaplan–Meier survival curves for all-cause mortality by diagnostic group. Legend: Kaplan–Meier curves depict the cumulative incidence of all-cause mortality among patients with myocardial infarction with obstructive coronary artery disease (MI-CAD), true myocardial infarction with non-obstructive coronary arteries (MINOCA), and myocardial infarction mimickers. True MINOCA patients exhibited similar long-term mortality to those with MI-CAD, whereas MINOCA mimickers had significantly lower mortality during follow-up. Statistical comparison was performed using the log-rank test.

**Figure 2 jcm-14-03006-f002:**
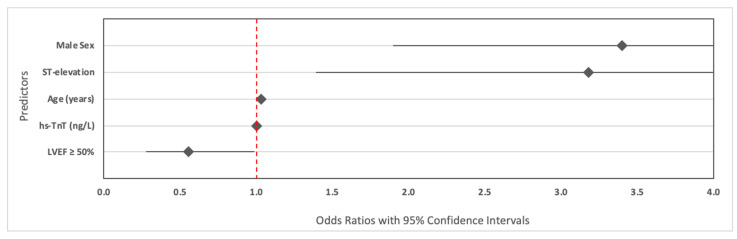
Forest plot of independent predictors of true MINOCA vs. MI-CAD. Legend: This forest plot illustrates the adjusted odds ratios (OR) and 95% confidence intervals (CI) from a multivariable logistic regression model identifying independent predictors of true myocardial infarction with non-obstructive coronary arteries (MINOCA) as compared to myocardial infarction with obstructive coronary artery disease (MI-CAD). Predictor variables included male sex, ST-elevation at admission (STEMI), age, high-sensitivity troponin T (hs-TnT) levels at admission, and left ventricular ejection fraction (LVEF ≥ 50%). An OR greater than 1 indicates increased likelihood of MI-CAD, while an OR less than 1 favors a diagnosis of MINOCA. The red vertical line at OR = 1.0 denotes the line of no effect.

**Table 1 jcm-14-03006-t001:** Baseline clinical characteristics of patients with MI-CAD, true MINOCA, and MINOCA mimickers.

	All (*n* = 1596)	True MINOCA (*n* = 117)	MI-CAD (*n* = 1359)	MINOCA Mimickers (*n* = 127)	*p* Value
**Age (years)**	64 (21)	63 (22)	64 (21)	38 (38)	<0.001
**Male**	1198 (75.1)	59 (53.2)	1053 (77.5)	86 (67.7)	<0.001
**Duration of hospitalization (days)**	64 (21)	6 (4)	7 (4)	6 (4)	<0.001
**Presentation**					
**STEMI**	808 (50.6)	6 (5.4)	801 (59)	1 (0.8)	<0.001
**NSTEMI**	619 (38.8)	61 (55)	558 (41.1)	0 (0)	<0.001
**Symptoms at Presentation**					
**Chest pain**	921 (57.7)	62 (55.9)	783 (57.7)	76 (59.8)	0.821
**Dyspnea**	128 (8)	14 (12.6)	100 (7.4)	14 (11)	0.063
**Fever**	40 (2.5)	0 (0)	4 (0.3)	36 (28.3)	<0.001
**Palpitations**	13 (0.8)	2 (1.8)	8 (0.6)	3 (2.4)	0.051
**Medical History**					
**Alcohol abuse**	27 (1.7)	1 (0.9)	22 (1.6)	4 (3.1)	0.353
**Smoking**	803 (50.3)	53 (47.7)	712 (52.4)	38 (29.9)	<0.001
**Arterial hypertension**	722 (45.2)	46 (41.4)	652 (48)	24 (18.9)	<0.001
**Dyslipidemia**	400 (25.1)	32 (28.8)	357 (26.3)	11 (8.7)	<0.001
**Atrial fibrillation**	122 (7.6)	7 (6.3)	107 (7.9)	8 (6.3)	0.701
**Diabetes mellitus**	390 (24.4)	22 (19.8)	352 (25.9)	16 (12.6)	0.002
**Thyroid disease**	92 (5.8)	10 (9)	75 (5.5)	7 (5.5)	0.315
**Lung disease**	64 (4)	2 (1.8)	53 (3.9)	9 (7.1)	0.102
**Chronic kidney disease**	109 (6.8)	1 (0.9)	107 (7.9)	1 (0.8)	<0.001
**Connective tissue disease**	28 (1.8)	4 (3.6)	19 (1.4)	5 (3.9)	0.035
**Valvular heart disease**	17 (1.1)	3 (2.7)	13 (1)	1 (0.8)	0.216
**Thoracic aneurysm**	17 (1.1)	1 (0.9)	16 (1.2)	0 (0)	0.458
**HFrEF**	242 (15.2)	4 (3.6)	225 (16.6)	13 (10.2)	<0.001
**HFmrEF**	303 (19)	13 (11.7)	283 (20.8)	7 (5.5)	<0.001
**HFpEF**	41 (2.6)	3 (2.7)	32 (2.4)	6 (4.7)	0.271
**Prior AMI**	150 (9.4)	1 (0.9)	146 (10.8)	3 (2.4)	<0.001
**Prior PCI or CABG**	247 (15.5)	3 (2.7)	241 (17.7)	3 (2.4)	<0.001
**Prior ischemic stroke**	15 (0.9)	3 (2.7)	11 (0.8)	1 (0.8)	0.137
**History of myocarditis**	7 (0.4)	0 (0)	0 (0)	7 (5.5)	<0.001
**History of pericarditis/pericardial effusion**	7 (0.4)	1 (0.9)	5 (0.4)	1 (0.8)	0.591
**ECG at Admission**					
**ST elevation**	585 (36.7)	12 (10.8)	539 (39.7)	34 (26.8)	<0.001
**ST depression**	328 (20.6)	14 (12.6)	307 (22.6)	7 (5.5)	<0.001
**Pathological Q waves**	260 (16.3)	14 (12.6)	240 (17.7)	6 (4.7)	<0.001
**Negative T waves**	330 (20.7)	44 (39.6)	258 (19)	28 (22)	<0.001
**Discharge Medication**					
**Antiplatelets**	1453 (91)	104 (93.7)	1325 (97.6)	24 (18.9)	<0.001
**DAPT**	1299 (81.4)	78 (70.3)	1219 (89.8)	2 (1.6)	<0.001
**Anticoagulants**	208 (13)	11 (9.9)	181 (13.3)	16 (12.6)	0.582
**Beta-blockers**	1393 (87.3)	83 (74.8)	1215 (89.5)	95 (74.8)	<0.001
**Calcium channel blockers**	219 (13.7)	27 (24.3)	179 (13.2)	13 (10.2)	0.002
**RAASi**	897 (56.2)	57 (51.4)	778 (57.3)	62 (48.8)	0.104
**MRA**	315 (19.7)	13 (11.7)	287 (21.1)	15 (11.8)	0.004
**Antilipidemic agent**	1400 (87.7)	98 (88.3)	1276 (94)	26 (20.5)	<0.001
**Statin only**	1302 (81.6)	78 (70.3)	1203 (88.6)	21 (16.5)	<0.001
**Statin + ezetimibe**	90 (5.6)	20 (18)	65 (4.8)	5 (3.9)	<0.001
**Management**					
**Thrombolysis**	98 (6.1)	2 (1.8)	96 (7.1)	0 (0)	<0.001
**Angiography**	1530 (95.9)	117 (100)	1359 (100)	54 (42.5)	<0.001
**PCI**	981 (61.5)	3 (2.7)	978 (72)	0 (0)	<0.001
**CMR**	145 (6.3)	75 (64.1)	9 (0.7)	61 (48)	<0.001
**CCTA**	75 (4.7)	65 (55.5)	3 (0.2)	7 (5.5)	<0.001
**CTPA**	16 (1)	0 (0)	11 (0.8)	5 (3.9)	0.002
**SPECT**	73 (4.6)	3 (2.7)	59 (4.3)	11 (8.7)	0.052
**Holter**	81 (5.1)	7 (6.3)	34 (2.5)	40 (31.5)	<0.001
**Admission Labs**					
**WBC** (×10^3^/μL)	9645 (4492)	8990 (4667)	9720 (4490)	8620 (4265)	0.005
**HGB** (g/dL)	13.5 (2.4)	13.4 (2.3)	13.5 (2.4)	13.8 (1.9)	0.595
**D-dimers** (ng/mL)	249 (309)	203 (185)	249 (308)	267 (292)	0.198
**Blood glucose** (mg/dL)	109 (47)	98 (37)	110 (47)	92 (20)	<0.001
**Creatinine** (mg/dL)	0.9 (0.33)	0.8 (0.3)	0.9 (0.3)	0.9 (0.3)	0.028
**CPK** (U/L)	249 (675)	129 (298)	260 (750)	142 (313)	<0.001
**CKMB** (U/L)	34 (57)	25 (26)	34 (58)	28 (29)	0.008
**hs-TnT** (ng/L)	701 (2464)	145 (444)	809 (2597)	207 (667)	<0.001
**CRP** (mg/L)	2.1 (8.4)	0.9 (3.2)	2.1 (8.5)	1.4 (19.2)	0.079
**NTproBNP** (pg/mL)	1176 (2235)	303 (1604)	1251 (2218)	281 (1356)	<0.001
**Total cholesterol** (mg/dL)	166 (62)	159 (67)	166 (62)	151 (47)	0.06
**TG** (mg/dL)	123 (77)	116 (79)	125 (78)	107 (60)	0.002
**HDL** (mg/dL)	38 (14)	46 (15)	38 (13)	38 (14)	<0.001
**LDL** (mg/dL)	97 (53)	90 (63)	98 (54)	90 (38)	0.105
**Peak hsTnT** (ng/L)	1143 (2871)	197 (699)	1314 (2963)	232 (766)	<0.001
**Echocardiography**					
**LVEF** (%)	53 (15)	59 (11)	52 (17)	60 (15)	<0.001
**LVEDd** (cm)	4.9 (0.9)	4.9 (0.6)	5 (0.9)	4.9 (0.8)	0.115
**E/E’**	7 (4)	6.8 (3)	7.2 (4)	6.3 (2)	0.003
**LA volume** (mL)	40 (18)	40 (18)	40 (18)	39 (18)	0.368
**Prognosis**					
**All-cause mortality**	427 (26.8)	18 (32.1)	405 (30.9)	4 (5.9)	<0.001

Legend: This table summarizes the demographic features, cardiovascular risk factors, clinical presentation, electrocardiographic, echocardiographic, laboratory findings, discharge medications, and mortality. Abbreviations are presented in detail in the main text. Values are expressed as median (interquartile range) or number (percentage), as appropriate.

**Table 2 jcm-14-03006-t002:** Independent predictors of MI-CAD vs. true MINOCA in multivariable logistic regression analysis.

Predictor Variables	Odds Ratio	Lower CI	Upper CI	*p* Value
Male sex (vs. female)	3.400	1.900	6.000	<0.001
ST-elevation at admission	3.180	1.388	7.284	0.006
Age (years)	1.033	1.011	1.054	0.002
hs-TnT (ng/L) at admission	1.005	1.002	1.008	0.004
LVEF ≥ 50% (vs. <50%)	0.555	0.280	0.991	0.046

Legend: This table presents the results of a multivariable logistic regression model aimed at identifying independent predictors of myocardial infarction with obstructive coronary artery disease (MI-CAD) in comparison to true myocardial infarction with non-obstructive coronary arteries (MINOCA). The dependent variable was diagnosis of MI-CAD vs. MINOCA. Odds ratios greater than 1 indicate increased likelihood of MI-CAD. Predictor variables included age, sex, ST-segment elevation at admission, high-sensitivity troponin T (hs-TnT, ng/L) at admission, and left ventricular ejection fraction (LVEF ≥ 50%). Odds ratios (OR), 95% confidence intervals (CI), and *p*-values are reported.

## Data Availability

The data presented in this study are available on request from the corresponding author.

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
