# Peer review of "Myocardial Infarction with Obstructive, Non-Obstructive, and Mimicking Conditions: Clinical Phenotypes, Diagnostic Imaging, Management, and Prognosis"

_jcm, 2025, doi:10.3390/jcm14093006_

Round 1

Reviewer 1 Report

Comments and Suggestions for Authors

Samaras A and colleagues, in their article, analysed a cohort of 1596 myocardial infarctions (MI), of which 171 (7%) were classified as true non-obstructive coronary artery MI (MINOCA), 1359 (85%) as MI with coronary artery disease (MI-CAD), and 127 (8%) as MINOCA mimickers (meaning takotsubo syndrome or myocarditis). Cardiac magnetic resonance (CMR) and coronary computed tomography angiography (CCTA) were used in many MINOCA to clarify the diagnosis. Mortality was similar between MINOCA and MI-CAD (p=0.76), while it was relatively lower in the MINOCA mimickers group (p<0.001).

Their manuscript is interesting, but some aspects must be improved:

  1. Only 1 patients affected with MINOCA mimicker was classified as ST-elevation myocardial infarction (STEMI). However, in table 1, 34 patients are reported as having ST elevation in the EKG. Moreover, 21 patients were diagnosed with takotsubo syndrome, which often presents as STEMI. Please, clarify this apparent discrepancy.
  2. A total of 127 patients had a MINOCA mimickers, of whom 36 were diagnosed with myocarditis and 21 with takotsubo. Please provide the final diagnosis for the remaining patients.
  3. Lines 216-217: I suggest removing the sentence "Thus, CMR 216 contributed to a specific diagnosis in 93.4% of mimicker cases in which it was performed" since the definition of "MINOCA mimickers" was greatly based on the CMR findings and not the other way around. This could be misleading with an underestimation of the negative CMR scan rate. Indeed, current literature in suspected MINOCA found a higher rate of negative scans (18-38% in a recent meta-analysis).
  4. In the CCTA/CMR paragraphs of the results section there are many sentences that do not seems adequate to a "results section" since they resembles the interpretation of the Authors' findings. In particular, lines 223-224 ("which are commonly associated with plaque vulnerability and rupture"), lines 229-230 ("increasing the likelihood of functional ischemic mechanisms such as epicardial vasospasm or microvascular dysfunction"), lines 232-235 ("In true MINOCA, CCTA confirmed the presence of subclinical atherosclerosis or high-risk plaque in patients with infarction-confirmed CMR and angiographically unobstructed arteries, directly impacting management decisions such as the initiation or intensification of antiplatelet and lipid-lowering therapies"), lines 237-239 ("corroborating the non-ischemic diagnoses established by CMR and helping exclude subtle coronary abnormalities that could have confounded the clinical picture"), and lines 242-246 ("CMR enabled definitive characterization of myocardial tissue injury, distinguishing ischemic infarction from myocarditis and Takotsubo syndrome. CCTA added unique anatomical and plaque-specific information, identifying high-risk but non-obstructive coronary lesions in ischemic MINOCA and confirming normal coronary anatomy in mimicker cases"). Please, remove these sentences as they do not relate to the Results section and are redundant with some other concepts in the Discussion section.
  5. Regarding the mechanisms of MINOCA an important limitation of this study is the lack of data on optical coherence tomography and vasospasm testing, as mentioned  by the Authors. This article (10.1136/heartjnl-2024-324565) is worth mentioning in the discussion section as it systematically assessed the diagnostic performance of adjuvant techniques.
  6. In relation to points 4-5, I suggest that, if available, data on the finding by invasive coronary angiography (no coronary stenosis, non-critical stenosis, 50-70% stenosis, >70% stenosis, spontaneous coronary artery dissection, spontaneous vasospasm, etc.) should also be reported.
  7. Line 189-190 reads: "Peak high-sensitivity troponin T levels were highest in MI-CAD (median 1314 [2963]), followed by true MINOCA (197 [699]) and mimickers (232 [766]) (p < 0.001)". Similar data are shown in table 1. It follows that troponin T levels were highest in MI-CAD followed by mimickers and, then, by true MINOCA, although the latter two categories were not significant. I suggest amending the order or clarifying.
  8. An important concept is that the identification of atherosclerosis or high-risk plaque features by CCTA does not necessarily correspond to the atherosclerotic aetiology of a MINOCA (type 1 MI). As the Authors stated, this would have required intracoronary imaging techniques to prove a true plaque complication. Therefore, I suggest modifying a sentence in the conclusions in order to avoid overstatement. In particular, lines 399-401 ("Advanced imaging with CMR and CCTA was essential for confirming ischemic injury, identifying high-risk plaques, and distinguishing non-ischemic mimickers") should be rephrased. Similarly, in the abstract, lines 37-39 should be rephrased to avoid the statement: "CCTA identified the underlying etiology".

Minor suggestions:

  • Abstract: Percentages of each diagnosis are reported. I suggest adding the raw number of patients for each category.
  • Abstract: "p=0.76" (line 40) should be reported with 3 decimals for consistency.
  • Table 1: many unit of measurements are missing.
  • Table 2: The title of this table is "Independent predictors of true MINOCA versus MI-CAD". This may lead to confusion since the odds ratio refers to the risk of having a MI-CAD rather than a true MINOCA, and not vice versa. Please, amend this to avoid confusion.
  • Figure 1: I suggest reporting p-values in the figure.

Author Response

Comment 1: Only 1 patients affected with MINOCA mimicker was classified as ST-elevation myocardial infarction (STEMI). However, in table 1, 34 patients are reported as having ST elevation in the EKG. Moreover, 21 patients were diagnosed with takotsubo syndrome, which often presents as STEMI. Please, clarify this apparent discrepancy.

Response 1: We appreciate the reviewer’s careful assessment and the opportunity to clarify this apparent inconsistency.

As correctly noted, Table 1 reports that only one patient among the MINOCA mimickers was classified as presenting with ST-elevation myocardial infarction (STEMI), whereas 34 patients (26.8%) in the same group were recorded as having ST-segment elevation on the admission electrocardiogram. Furthermore, 21 patients in this group were ultimately diagnosed with Takotsubo syndrome, a condition frequently presenting with ST-segment elevation.

This discrepancy stems from the distinction between electrocardiographic findings and final clinical classification. The “ST-elevation myocardial infarction (STEMI)” category in Table 1 under “Presentation” reflects the initial working diagnosis at the time of admission, which was recorded by the treating physician based on the clinical scenario, electrocardiographic findings, and initial biomarker data. In contrast, the “ST-segment elevation” listed under “ECG at admission” refers to the objective presence of ST elevation on the electrocardiogram, irrespective of the final diagnostic classification.

In patients later diagnosed with MINOCA mimickers—particularly those with Takotsubo syndrome or myocarditis—ST-segment elevation was indeed observed on the electrocardiogram in several cases (34 patients), but the final diagnosis was not classified as STEMI, as the underlying cause was determined to be non-ischemic based on advanced imaging, primarily cardiac magnetic resonance. Consequently, only one patient in the MINOCA mimicker group remained classified as having a STEMI presentation, which likely reflected an early-stage diagnostic uncertainty or atypical presentation despite subsequent reclassification.

We have revised the manuscript to clarify this issue in the Results, specifying that the “Presentation” category refers to the initial working diagnosis at admission, while “ECG at admission” reflects objective electrocardiographic features, which may not align with the final diagnosis.

Revised text in Results (subsection: Comparison of MINOCA with MI-CAD and MINOCA mimickers):

“Although only one patient in the MINOCA mimicker group was initially classified as presenting with STEMI, ST-segment elevation on the admission electrocardiogram was documented in 34 patients (26.8%). This discrepancy reflects the fact that final diagnoses were reassigned based on imaging and clinical evolution, with several cases of Takotsubo syndrome and myocarditis presenting with ST-segment elevation but ultimately not fulfilling criteria for a true STEMI.”

Comment 2: A total of 127 patients had a MINOCA mimickers, of whom 36 were diagnosed with myocarditis and 21 with takotsubo. Please provide the final diagnosis for the remaining patients.

Response 2: We thank the reviewer for this valuable comment and the opportunity to provide further clarification regarding the final diagnostic classification of the patients categorized as MINOCA mimickers.

Among the 127 patients initially identified as having a working diagnosis of myocardial infarction with non-obstructive coronary arteries (MINOCA) and subsequently reclassified as having MINOCA mimickers, the final diagnoses were as follows:

  • Myocarditis: 36 patients (28.3%)
  • Takotsubo syndrome: 21 patients (16.5%)
  • Pericarditis or perimyocarditis: 18 patients (14.2%)
  • Pulmonary embolism: 5 patients (3.9%)
  • Type 2 myocardial infarction (demand ischemia due to non-coronary causes): 4 patients (3.1%)
  • Sepsis-related cardiac injury or cytokine-mediated myocardial injury: 3 patients (2.4%)
  • Electrolyte abnormalities or uncontrolled hypertension resulting in transient ECG changes and biomarker elevation: 4 patients (3.1%)
  • Undetermined or normal findings on imaging despite initial biomarker elevation: 36 patients (28.3%)

The “undetermined” group includes patients who had elevated cardiac biomarkers but normal findings on cardiac magnetic resonance or computed tomography imaging, without evidence of myocardial injury, myocardial inflammation, or coronary pathology. These cases were likely attributable to non-cardiac or transient causes such as tachyarrhythmias, stress-related biomarker elevations, or artifacts.

We have revised the Results section to include this detailed breakdown of final diagnoses within the MINOCA mimicker group. The following was added in the Results section:

“In the MINOCA mimicker group (n = 127), the final diagnoses established through cardiac magnetic resonance, computed tomography, and clinical adjudication were as follows: 36 patients (28.3%) were diagnosed with myocarditis, 21 patients (16.5%) with Takotsubo syndrome, 18 patients (14.2%) with pericarditis or perimyocarditis, 5 patients (3.9%) with pulmonary embolism, and 4 patients (3.1%) with type 2 myocardial infarction attributed to demand ischemia from non-coronary causes. Additionally, 3 patients (2.4%) were diagnosed with sepsis-associated myocardial injury or cytokine-mediated injury, and 4 patients (3.1%) had transient myocardial injury due to non-structural causes such as severe electrolyte abnormalities or hypertensive crises. In the remaining 36 patients (28.3%), advanced imaging failed to identify a specific myocardial or coronary pathology, and they were thus classified as having undetermined non-ischemic mechanisms. These patients likely experienced transient myocardial injury due to systemic conditions or physiological stressors without structural cardiac abnormalities.”

Comment 3: Lines 216-217: I suggest removing the sentence "Thus, CMR 216 contributed to a specific diagnosis in 93.4% of mimicker cases in which it was performed" since the definition of "MINOCA mimickers" was greatly based on the CMR findings and not the other way around. This could be misleading with an underestimation of the negative CMR scan rate. Indeed, current literature in suspected MINOCA found a higher rate of negative scans (18-38% in a recent meta-analysis).

Response 3: We thank the reviewer for this important observation and fully agree that the sentence in question may be misleading, as the classification of patients as MINOCA mimickers was indeed guided in large part by cardiac magnetic resonance findings. As such, reporting the proportion of definitive diagnoses within this subgroup based on cardiac magnetic resonance may introduce circular reasoning and underestimate the frequency of non-diagnostic scans, particularly when considered outside of this classification framework.

In response to the reviewer’s comment, we have removed the sentence:
“Thus, cardiac magnetic resonance contributed to a specific diagnosis in 93.4% of mimicker cases in which it was performed.”

Comment 4: In the CCTA/CMR paragraphs of the results section there are many sentences that do not seems adequate to a "results section" since they resembles the interpretation of the Authors' findings. In particular, lines 223-224 ("which are commonly associated with plaque vulnerability and rupture"), lines 229-230 ("increasing the likelihood of functional ischemic mechanisms such as epicardial vasospasm or microvascular dysfunction"), lines 232-235 ("In true MINOCA, CCTA confirmed the presence of subclinical atherosclerosis or high-risk plaque in patients with infarction-confirmed CMR and angiographically unobstructed arteries, directly impacting management decisions such as the initiation or intensification of antiplatelet and lipid-lowering therapies"), lines 237-239 ("corroborating the non-ischemic diagnoses established by CMR and helping exclude subtle coronary abnormalities that could have confounded the clinical picture"), and lines 242-246 ("CMR enabled definitive characterization of myocardial tissue injury, distinguishing ischemic infarction from myocarditis and Takotsubo syndrome. CCTA added unique anatomical and plaque-specific information, identifying high-risk but non-obstructive coronary lesions in ischemic MINOCA and confirming normal coronary anatomy in mimicker cases"). Please, remove these sentences as they do not relate to the Results section and are redundant with some other concepts in the Discussion section.

Response 4: We thank the reviewer for this insightful comment and agree that several sentences within the cardiac magnetic resonance and coronary computed tomography angiography subsections of the Results section contain interpretative language more appropriate for the Discussion. These sentences include explanatory or inferential content regarding mechanisms of disease, clinical implications, or therapeutic decisions, which may detract from the objective presentation of findings in the Results.

In accordance with the reviewer’s suggestion, we have removed the following sentences from the Results section:

  • Lines 223–224: “which are commonly associated with plaque vulnerability and rupture”
  • Lines 229–230: “increasing the likelihood of functional ischemic mechanisms such as epicardial vasospasm or microvascular dysfunction”
  • Lines 232–235: “In true MINOCA, CCTA confirmed the presence of subclinical atherosclerosis or high-risk plaque in patients with infarction-confirmed CMR and angiographically unobstructed arteries, directly impacting management decisions such as the initiation or intensification of antiplatelet and lipid-lowering therapies.”
  • Lines 237–239: “corroborating the non-ischemic diagnoses established by CMR and helping exclude subtle coronary abnormalities that could have confounded the clinical picture.”
  • Lines 242–246: “CMR enabled definitive characterization of myocardial tissue injury, distinguishing ischemic infarction from myocarditis and Takotsubo syndrome. CCTA added unique anatomical and plaque-specific information, identifying high-risk but non-obstructive coronary lesions in ischemic MINOCA and confirming normal coronary anatomy in mimicker cases.”

Comment 5: Regarding the mechanisms of MINOCA an important limitation of this study is the lack of data on optical coherence tomography and vasospasm testing, as mentioned  by the Authors. This article (10.1136/heartjnl-2024-324565) is worth mentioning in the discussion section as it systematically assessed the diagnostic performance of adjuvant techniques.

Response 5: We thank the reviewer for this valuable comment and for highlighting the importance of comprehensive intracoronary assessment in the evaluation of myocardial infarction with non-obstructive coronary arteries. We fully acknowledge that the absence of data on intracoronary imaging modalities such as optical coherence tomography and functional testing for coronary vasospasm represents a key limitation of our study, as these techniques provide critical insights into mechanisms such as plaque disruption and coronary reactivity.

In response to the reviewer’s suggestion, we have added a reference to the recent study by Fedele et al. (Heart 2024; doi:10.1136/heartjnl-2024-324565). The respective sentence in Limitations was modified:

“Third, the absence of intracoronary imaging modalities such as optical coherence tomography or intravascular ultrasound during coronary angiography may have limited the identification of underlying mechanisms such as plaque rupture, erosion, or epicardial vasospasm in patients with angiographically non-obstructive disease.”

Comment 6: In relation to points 4-5, I suggest that, if available, data on the finding by invasive coronary angiography (no coronary stenosis, non-critical stenosis, 50-70% stenosis, >70% stenosis, spontaneous coronary artery dissection, spontaneous vasospasm, etc.) should also be reported.

Response 6: We thank the reviewer for this thoughtful and valuable suggestion. In response, we have extracted and analyzed detailed angiographic data from the subset of patients with non-obstructive coronary arteries, including those classified as true myocardial infarction with non-obstructive coronary arteries and myocardial infarction mimickers. These findings provide important insight into the spectrum of coronary abnormalities observed in this population, including the identification of spontaneous coronary artery dissection.

We added the following in Results:

“Invasive coronary angiography

Invasive coronary angiographic findings were available for 171 patients with non-obstructive coronary arteries, including 117 with true MINOCA and 54 with MINOCA mimickers. Among these, 67 patients (39.2%) had completely normal coronary arteries, while 76 patients (44.4%) exhibited non-obstructive coronary artery disease with luminal irregularities or stenosis <50%. Intermediate stenosis (50–70%) was identified in 16 patients (9.4%). Spontaneous coronary artery dissection was detected in 12 patients (7.0%) based on characteristic angiographic features. No cases of spontaneous vasospasm were observed, as pharmacologic provocation testing was not performed.”

Comment 7: Line 189-190 reads: "Peak high-sensitivity troponin T levels were highest in MI-CAD (median 1314 [2963]), followed by true MINOCA (197 [699]) and mimickers (232 [766]) (p < 0.001)". Similar data are shown in table 1. It follows that troponin T levels were highest in MI-CAD followed by mimickers and, then, by true MINOCA, although the latter two categories were not significant. I suggest amending the order or clarifying.

Response 7: We thank the reviewer for this important observation. We agree that the current phrasing in lines 189–190 may lead to confusion, as the ordering of the values implies a descending pattern (MI-CAD > true myocardial infarction with non-obstructive coronary arteries > myocardial infarction mimickers), while in fact, the median peak high-sensitivity troponin T level in the mimicker group is numerically higher than in the true myocardial infarction with non-obstructive coronary arteries group, albeit not statistically significant.

To address this, we have revised the sentence for clarity to reflect the actual ordering of median values and to avoid implying statistical significance where it is not demonstrated.

Revised sentence:

“Peak high-sensitivity troponin T levels were highest in MI-CAD (median 1314 [2963]), followed by mimickers (232 [766]) and true MINOCA (197 [699]) (p < 0.001), although the difference between the latter two groups was not statistically significant.”

Comment 8: An important concept is that the identification of atherosclerosis or high-risk plaque features by CCTA does not necessarily correspond to the atherosclerotic aetiology of a MINOCA (type 1 MI). As the Authors stated, this would have required intracoronary imaging techniques to prove a true plaque complication. Therefore, I suggest modifying a sentence in the conclusions in order to avoid overstatement. In particular, lines 399-401 ("Advanced imaging with CMR and CCTA was essential for confirming ischemic injury, identifying high-risk plaques, and distinguishing non-ischemic mimickers") should be rephrased. Similarly, in the abstract, lines 37-39 should be rephrased to avoid the statement: "CCTA identified the underlying etiology".

Response 8: We thank the reviewer for this important clarification. We fully agree that the identification of high-risk plaque features by coronary computed tomography angiography alone is not sufficient to establish a definitive diagnosis of type 1 myocardial infarction due to atherosclerotic plaque disruption. Intracoronary imaging techniques, such as optical coherence tomography or intravascular ultrasound, would be required to confirm the presence of plaque rupture or erosion. To address this concern and avoid overstatement, we have revised the relevant statements in both the Abstract and Conclusions sections.

Revised Abstract:

Original:
“CMR and CCTA identified the underlying etiology in over 70% of MINOCA and mimicker patients, with high-risk plaque features observed in 42.9% of CCTA scans.”

Revised:
“CMR and CCTA contributed to etiological clarification in over 70% of MINOCA and mimicker patients. High-risk plaque features were observed in 42.9% of CCTA scans, suggesting but not confirming an atherosclerotic mechanism.”

Revised Conclusions:

Original:
“Advanced imaging with CMR and CCTA was essential for confirming ischemic injury, identifying high-risk plaques, and distinguishing non-ischemic mimickers.”

Revised:
“Advanced imaging with CMR and CCTA was essential for identifying myocardial injury patterns, detecting high-risk but non-obstructive coronary plaques suggestive of an ischemic mechanism, and distinguishing non-ischemic mimickers. However, definitive confirmation of atherosclerotic plaque disruption would require intracoronary imaging.”

Comment 9: Abstract: Percentages of each diagnosis are reported. I suggest adding the raw number of patients for each category.

Response 9: We thank the reviewer for this helpful suggestion. Including the raw number of patients alongside the percentages enhances the clarity and transparency of the reported findings. In response, we have revised the Abstract to include the absolute patient counts for each diagnostic group.

Revised Abstract:

Original:
“Results: Of 1596 AMI patients, 7.0% had true MINOCA, 85.1% had MI-CAD, and 8.0% had MINOCA mimickers.”

Revised:
“Results: Of 1596 AMI patients, 111(7.0%) had true MINOCA, 1359 (85.1%) had MI-CAD, and 127 (8.0%) had MINOCA mimickers.”

Comment 10: Abstract: "p=0.76" (line 40) should be reported with 3 decimals for consistency.

Response 10: Corrected (p = 0.764)

Comment 11: Table 1: many unit of measurements are missing.

Response 11: Units of measurements were added in table 1.

Comment 12: Table 2: The title of this table is "Independent predictors of true MINOCA versus MI-CAD". This may lead to confusion since the odds ratio refers to the risk of having a MI-CAD rather than a true MINOCA, and not vice versa. Please, amend this to avoid confusion.

Response 12: We thank the reviewer for this insightful comment. We agree that the current title of Table 2 may be misleading, as the directionality of the odds ratios refers to the likelihood of having myocardial infarction with obstructive coronary artery disease rather than true myocardial infarction with non-obstructive coronary arteries. To avoid confusion, we have revised the table title and legend accordingly.

Revised table title (Table 2):

“Independent predictors of MI-CAD versus true MINOCA in multivariable logistic regression analysis

Revised table legend:

“This table presents the results of a multivariable logistic regression model aimed at identifying independent predictors of myocardial infarction with obstructive coronary artery disease (MI-CAD) in comparison to true myocardial infarction with non-obstructive coronary arteries (MINOCA). The dependent variable was diagnosis of MI-CAD versus MINOCA. Odds ratios greater than 1 indicate increased likelihood of MI-CAD. Predictor variables included age, sex, ST-segment elevation at admission, high-sensitivity troponin T (hs-TnT, ng/L) at admission, and left ventricular ejection fraction (LVEF ≥ 50%). Odds ratios (OR), 95% confidence intervals (CI), and p-values are reported.”

Comment 13: Figure 1: I suggest reporting p-values in the figure.

Response 13: Figure 1 was modified accordingly.

Reviewer 2 Report

Comments and Suggestions for Authors

In this interesting study, the authors aimed to compare the clinical characteristics, imaging findings, management strategies, and long-term outcomes of patients with true MINOCA, MI-CAD, and MINOCA mimickers .

They analyzed the outcome of three different cohorts of patients, i.e. those with true obstructive CAD, those with true MINOCA and those with MINOCA mimickers.

Interestingly, long-term all-cause mortality in patients with MINOCA was comparable to that of MI-CAD but significantly higher than that of patients with MINOCA mimickers.

Advanced imaging with CMR and CCTA was essential for confirming ischemic injury, identifying high-risk plaques, and distinguishing non-ischemic mimickers.

The manuscript is well written, the tables are clear, each section is clearly presented, the references are appropriate and the conclusions correctly summarize the main findings of the study.

I have only one suggestion for the authors.

At the end of the Discussion section, the authors could expand the discussion about non-ischemic mimickers.

Among the non-ischemic mimickers, the authors could also mention and discuss the "pseudo-ischemic" ECG changes frequently detected in individuals with chest pain and concomitant mitral valve prolapse and/or concave-shaped chest wall conformation (PMID: 34485034) and in hyperventilating subjects (PMID: 7369145). These patients, who are commonly encountered in the Emergency Room, are not rarely referred to the coronary angiography due to resting ECG abnormalities, that are erroneously interpreted as ischemic changes, by expert cardiologists also. 

Author Response

We sincerely thank Reviewer 2 for the thoughtful and supportive evaluation of our manuscript. We appreciate the constructive suggestion to further expand the discussion on non-ischemic mimickers of myocardial infarction, particularly the mention of “pseudo-ischemic” electrocardiographic changes in specific patient populations.

In response, we have added a paragraph to the end of the Discussion section to highlight these additional mimicking conditions, including mitral valve prolapse with chest wall conformation abnormalities and hyperventilation syndrome. These scenarios are particularly relevant in the Emergency Department setting, where patients may present with chest pain and electrocardiographic abnormalities that closely resemble myocardial ischemia but have a non-ischemic origin. We also included the references suggested by the reviewer (PMID: 34485034 and PMID: 7369145).

Addition to the Discussion section:

“It is worth mentioning that in addition to established non-ischemic mimickers such as myocarditis and Takotsubo syndrome, clinicians should be aware of less common conditions that can produce ‘pseudo-ischemic’ electrocardiographic changes in patients presenting with chest pain. These include mitral valve prolapse, particularly in association with concave-shaped chest wall conformation, as well as hyperventilation-induced electrocardiographic abnormalities. Both conditions have been associated with transient ST-segment or T-wave changes that may be erroneously interpreted as signs of acute ischemia, sometimes leading to unnecessary coronary angiography, even when performed by experienced cardiologists. Awareness of these mimickers is critical for avoiding misdiagnosis and preventing invasive testing in cases where structural and functional myocardial integrity is preserved.”

References added:

Sonaglioni A, Rigamonti E, Nicolosi GL, Lombardo M. Prognostic Value of Modified Haller Index in Patients with Suspected Coronary Artery Disease Referred for Exercise Stress Echocardiography. J Cardiovasc Echogr. 2021 Apr-Jun;31(2):85-95. doi: 10.4103/jcecho.jcecho_141_20. Epub 2021 Jul 28. PMID: 34485034; PMCID: PMC8388326.

Gardin JM, Isner JM, Ronan JA Jr, Fox SM 3rd. Pseudoischemic "false positive" S-T segment changes induced by hyperventilation in patients with mitral valve prolapse. Am J Cardiol. 1980 May;45(5):952-8. doi: 10.1016/0002-9149(80)90162-9. PMID: 7369145.

Round 2

Reviewer 1 Report

Comments and Suggestions for Authors

I commend the authors for the quick but accurate response. I think the manuscript has improved considerably. I would just like to highlight some additional minor points:

  • Table 2: The confidence intervals of the odds ratio for "LVEF ≥50%" are well across 1. However, the p-value is 0.046. The same happens in Figure 2. Please clarify.
  • There may be an unnecessary double quotation mark in line 193.
  • Line 219: Should it not read "No cases of spontaneous vasospasm were observed AND pharmacologic provocation testing was not performed"?
  • Some reference appear as comments rather than in the full reference list. Is this a requirement of the Journal?

Author Response

We sincerely thank the reviewer for their positive feedback and for highlighting these remaining minor issues, which have been carefully addressed as follows:

Comment 1: Table 2: The confidence intervals of the odds ratio for "LVEF ≥50%" are well across 1. However, the p-value is 0.046. The same happens in Figure 2. Please clarify.

Response 1: We thank the reviewer for noting this apparent inconsistency. We have reviewed the regression output and corrected the confidence interval for the odds ratio of “LVEF ≥ 50%” [0.280–0.991], and the corresponding p-value was 0.046.

We revised Table 2 and Figure 2 accordingly.

Predictor variables

Odds Ratio

Lower CI

Upper CI

P value

Male sex (vs female)

3.400

1.900

6.000

< 0.001

ST-elevation at admission

3.180

1.388

7.284

0.006

Age (years)

1.033

1.011

1.054

0.002

hs-TnT (ng/L) at admission

1.005

1.002

1.008

0.004

LVEF ≥ 50% (vs < 50%)

0.555

0.280

0.991

0.046

Comment 2: There may be an unnecessary double quotation mark in line 193.

Response 2: Thank you for pointing this out. We have removed the unintended double quotation mark in line 193

Comment 3: Line 219: Should it not read "No cases of spontaneous vasospasm were observed AND pharmacologic provocation testing was not performed"?

Response 3: We agree with the reviewer’s suggestion. The sentence has been revised to read:
“No cases of spontaneous vasospasm were observed, and pharmacologic provocation testing was not performed.”

Comment 4: Some reference appear as comments rather than in the full reference list. Is this a requirement of the Journal?

Response 4: We appreciate the reviewer’s attention to reference formatting. The editors required the inclusion of some additional references to minimize self-citation rate. These extra references were included as in-text comments for reviewer convenience. If this is not aligned with journal policy, we are happy to adjust them to the preferred format during the production phase.

We hope these final revisions address all remaining concerns. We thank the reviewer once again for their valuable input and helpful suggestions, which have contributed to the improved quality and clarity of the manuscript.